# Does Replacing Maize with Barley Affect the Animal Performance and Rumen Fermentation, including Methane Production, of Beef Cattle Fed High-Concentrate Diets On-Farm?

**DOI:** 10.3390/ani13193016

**Published:** 2023-09-25

**Authors:** Amira Arbaoui, Antonio de Vega

**Affiliations:** Departamento de Producción Animal y Ciencia de los Alimentos, Instituto Agroalimentario de Aragón (IA2), Universidad de Zaragoza-CITA, Miguel Servet 177, 50013 Zaragoza, Spain; arbaoui.amira@yahoo.fr

**Keywords:** beef cattle, cereal type, performance, rumen fermentation, methane

## Abstract

**Simple Summary:**

The mitigation of greenhouse gas (GHG) emissions is a must in order to fight the climate emergency. Ruminants are responsible for over 75% of GHG emissions from livestock, with enteric methane being of paramount importance. Animals consuming high-concentrate diets produce less enteric methane than those fed high-forage diets, but the predominant grain in the concentrate may have an influence. On these grounds, the objective of this study was to assess, in commercial farm conditions, the effects of a partial substitution of maize with barley on animal performance and rumen fermentation in intensively reared beef calves. The predominant grain in the concentrate had no effect, in general, on animal performance or rumen fermentation. The general population and environmentalists should then be aware that a partial substitution of maize with barley in the concentrate offered to beef calves does not seem a promising strategy to decrease the emissions of enteric methane on-farm.

**Abstract:**

Ruminants fed high-concentrate diets produce less enteric methane than those fed high-forage diets, but not all grains are equally effective in reducing methane production. This study aimed to examine, in farm conditions, the effects of a partial substitution of maize with barley on animal performance and rumen fermentation, including methane production, of intensively reared beef calves (ca. 0.9:0.1 concentrate to forage ratio). Ninety-six beef calves were fed a concentrate with 45.5% maize and 15% barley (n = 48; M) or a concentrate with 15.5% maize and 45% barley (n = 48; B). Both the concentrate and barley straw were offered ad libitum. The type of concentrate did not have a significant effect (*p* > 0.05) on final live weight, average daily gain, carcass dressing percentage or intake of concentrate and straw. Dry matter and organic matter digestibility were higher (*p* < 0.05) for the M (75.4% and 76.6%) than for the B (71.0% and 73.1%) treatment, but with no effect on digestible organic matter intake. In general, the majority cereal in the concentrate did not affect rumen fermentation, including methane production, or the degradability of dry matter and starch. A partial substitution of maize with barley in the concentrate offered to beef calves does not seem a promising strategy to decrease the emissions of enteric methane on-farm.

## 1. Introduction

Enteric emissions of greenhouse gases (GHG) by ruminants (mainly beef cattle) are of major concern, especially due to their role in global warming [1,2]. Among GHG, methane contributes most, in terms of CO_2_ equivalents, to warming potential [3]. Another negative implication of enteric methane production by ruminants is a loss of 2–12% of gross energy intake [4,5]. In order to minimize those effects, ruminal fermentation has been greatly manipulated over several decades by increasing the amount of concentrate feed in the diet to increase the effectiveness of nutrient utilization, improving animal productivity and reducing environmental impacts [6,7]. However, this effect is not linear, and Sauvant et al. [8] have stated that the inclusion of grain in the diets of dairy cows has a limited effect on methane yield until the proportion of concentrate in the diet is greater than 30%.

High-starch rations (containing great proportions of, e.g., wheat, barley or maize) favour propionate production, creating an alternative hydrogen sink to methanogenesis [9] and hence decreasing the ratio of methane/organic matter fermented in the rumen [10]. However, starch is rapidly fermented in the rumen, resulting in high VFA and/or lactate concentrations which contribute to an increased risk of acute or sub-acute ruminal acidosis (SARA) [11]. In these conditions, the growth of rumen methanogens is inhibited [12] and protozoal numbers are decreased, limiting the transfer of hydrogen from protozoa to methanogens [13].

Not all grains are equally effective in reducing methane production in the rumen, and there are indications that the response of methane production to different proportions of grains may not be linear [14]. Moate et al. [15] found that dairy cows produced less methane when fed wheat than when fed maize or barley, and Herrera-Saldana et al. [16] observed in vitro that maize grain has a lower rate and extent of degradation than barley grain. This has also been confirmed in sacco [17,18], where dry matter (DM) and starch from barley had a faster degradation rate than DM and starch from maize. A faster rate of rumen degradation of barley would decrease pH in the compartment, damaging the ability of methanogens to produce methane. On the other hand, Fellner et al. [19] recorded in vitro a higher methane production when fermenting barley than maize.

A few reasons may account for the discrepancy between papers on methane production from different grain sources: 1. When working in vitro, grains should be incubated in ruminal fluid from animals adapted to that specific grain type [14]; in the case of Fellner et al. [19], the donor animal (a dairy cow) was fed a forage diet. 2. Diet ingredients (including forage) may have different fibre content, affecting the ruminal fluid pH and hence methanogenic capacity in vivo [15]. 3. Energy intake/growth rate also influence methane production in vivo [20]. 4. Methane production obtained in vitro using pure substrates may not represent the in vivo facts with more complex diets [14], indicating the need for caution when interpreting the results from in vitro studies.

Even though many papers have been published on the effects of replacing maize with barley on animal performance and the microbial fermentation of beef cattle fed high-concentrate rations [21,22], far fewer have dealt with methane production in vivo [23,24] and, to our knowledge, none in commercial farm conditions.

As methane emissions in vivo seem to depend on diet characteristics and energy intake/growth rate, we hypothesized that (1) partially substituting maize with barley in the diet of intensively reared beef cattle may affect the methane yield; (2) this substitution will not increase the acidosis risk; and (3) this substitution will not affect animal performance. Hence, the objective of the present work was to assess the on-farm effects of partially substituting maize with barley on the animal performance and rumen fermentation, including methane production, of intensively-reared beef cattle.

## 2. Materials and Methods

### 2.1. Animals and Diets

The experiment lasted 94 days, and was carried out entirely at the commercial farm Murilló Fresh Foods SL, in the Spanish Pyrenees. Ninety-six Montbéliarde crossbred male calves were distributed, according to their weight and age, in two pens of forty-eight animals each, corresponding to two treatments: a control compound feed (M) made up of maize as the majority cereal (45.5% maize and 15% barley), and a treatment (B) in which the percentages of barley and maize were exchanged (15.5% maize and 45% barley). All concentrates were made up at the compound feed factory at the farm. Animals were assigned to both diets in order to obtain similar average weights and ages for the two groups (314 ± 3.1 kg and 209 ± 1.6 days for treatment M, and 313 ± 3.1 kg and 207 ± 1.5 days for treatment B).

Six animals from each treatment (those with a weight closer to the average of each group) were fitted with a 150 mm long, 15 mm internal diameter (i.d.) permanent cannula in the dorsal sac of the rumen. Another two animals per treatment were cannulated with a semi-rigid cannula 95 mm i.d. and 200 mm long. Animal care, handling and surgical procedures were approved by the Ethics Committee of the University of Zaragoza (ethical approval code PI26_21). The care and management of animals were performed according to the Spanish Policy for Animal Protection RD 1201/05, which meets the EU Directive 86/609 on the protection of animals used for experimental and other scientific purposes.

Compound feeds (Table 1) were formulated to be isoenergetic (2777 kcal of metabolizable energy (ME)/kg) and isoproteic (15% protein), and were offered ad libitum to the animals through a continuous feeding system. Barley straw, also offered ad libitum, was used as a forage source. The calves had free access to water from the supply network throughout the experimental period. The protein content of the concentrates was slightly higher than expected from the formulae. In addition, concentrate M presented a higher content in EE and starch, and lower in NDF and ADF, than concentrate B, the last three results being attributable to the presence of husks in barley.

### 2.2. Experimental Procedures

Animals were weighed every two weeks, estimating the average daily gain (ADG) as the regression coefficient of individual live weight (LW) on time. Approximately halfway through the experimental period, digestibility balances were carried out in the animals fitted with the small cannula. Due to limitations in the availability of metabolism cages (four; 1.4 m width, 2.3 m high, and 3.2 m length), three consecutive balances were carried out with two animals from each treatment per balance. Each balance lasted eight days, three for adaptation to the metabolism cages and five for data collection. During balances, concentrates and straw were offered once daily (9:00) in separate troughs, and at the same time representative samples were taken from concentrates and straw offered, and from the residues left by the animals the previous day. The daily amount of both concentrates and straw was adjusted to ensure at least 10% refusals. Samples of feed components and refusals were pooled according to balance and analysed for chemical composition (Table 1). Representative samples of offered feeds and refusals were dried at 104 °C for 24 h, to determine individual dry matter intake (DMI). Total faeces were collected daily from each animal, and a 5% aliquot taken, frozen at −20 °C, pooled by balance and freeze-dried. *n*-Alkane recovery was also assessed during digestibility balances.

On day 4 and day 6 of each digestibility balance, rumen gas samples (in duplicate) were obtained at 0 h (before feeding), and at 3, 6 and 9 h after feeding. To this purpose, a silicon tube (0.3 mm i.d.) was inserted about 10 cm through the cannula, which was sealed with plasticine to avoid the entrance of air into the rumen. The outer end of the tube was submerged in a plastic container with water, and once the air was bubbling a sample (1 mL) of the gas in the tube was obtained with a syringe and immediately transferred to a chromatography vial that had previously been vacuum-pressured. Right after the gas sampling and withdrawal of the silicon tube, about 200 mL rumen fluid was removed from the rumen of each animal for each sampling time, using a customized vacuum device connected to a 0.6 cm i.d. semi-rigid tube with 2 mm pores. Representative samples were taken, moving the tube in all directions inside the rumen while sampling. Then, rumen fluid was strained through a 1 mm pore size metal mesh sieve, and the pH was immediately measured using a portable pH meter (model Seven2GO, Mettler-Toledo AG, Schwerzenbach, Switzerland). Rumen fluid aliquots were taken, in duplicate, for ammonia, lactic acid and volatile fatty acids (VFA) analysis, and were analysed following the procedures described by Gimeno et al. [21].

As an indirect method to assess gas production from the animals fed the different dietary treatments, an in vitro incubation of the two rations (including concentrate M or B and straw, in the same proportions observed during the digestibility balances) was carried out, following the methods described by Theodorou et al. [25] and modified by Amanzougarene and Fondevila [26] for high-concentrate diets. Each mixture of straw and concentrate from each digestibility balance (two treatments × three balances) was incubated in 116 mL glass bottles. The rumen inoculum was obtained from the cannulated animals assigned to each treatment, and hence the experimental unit was the animal. Nine replicas were incubated per animal, making a total of 108 bottles, which were incubated in a single run. Three additional bottles without substrate were included as blanks. Pressure was recorded at 2, 4, 6, 8, 10, 12, 16 and 24 h in the blanks and in 6 out of the 9 replicates per treatment and balance, by means of an HD 2124.02 manometer fitted with a TP804 pressure gauge (Delta Ohm, Caselle di Salvazzano, Italy). In the 3 remaining replicas, pressure was recorded at 6, 12 and 24 h, withdrawing gas samples for methane analysis, using vacuum-pressured chromatography vials, at 6 and 12 h. Pressure was converted into volume using a pre-stablished linear regression equation between pressure and know air volumes [27]. The gas volume recorded for each incubation time was estimated as the average of all replicas of the same treatment and balance, expressed per unit of incubated organic matter (OM).

Once the digestibility balances were finished, concentrates were incubated in sacco in the rumen of cattle fitted with large cannulae (each concentrate in the rumen of the animals consuming it). Approximately 5 g samples of concentrates were weighed in polyester bags (10 × 16 cm; pore size 48 µm; Sefar Maissa S.A.U., Cardedeu (Barcelona, Spain)), which were incubated in duplicate (two bags per animal and hour) for 3, 6, 9, 24, 32, 50 and 56 h. After extraction, bags were rinsed with tap water and frozen at −20 °C. Once all bags had been withdrawn, they were washed for 30 min in a semiautomatic small washing machine (Jata S.A., (Tudela, Spain)). Two additional bags per concentrate, containing non-incubated material, were also washed to determine the soluble fraction. After washing, bags were dried at 60 °C for 48 h, and the residues inside stored in plastic containers until analysis.

The rate of passage of the two concentrates was assessed after the bags’ incubation by introducing 150 g of Yb-labelled concentrates [28] through the cannula of the animals consuming them, and through faecal sampling, directly from the rectum, at 17, 26, 41 and 50 h after marker dosage. Faecal samples were frozen at −20 °C until marker analysis.

At the time of the animals’ next weighing after the rate of passage assessment, a faecal sample was obtained from the animals fitted with the small cannula (those who were used for digestibility balance) to analyse the concentration of *n*-alkanes. As the individual recovery of these markers was calculated during the digestibility balances, the objective was to try to obtain intake and digestibility values in the group- and loose-housed animals.

All animals were slaughtered at a target LW of 500 kg, and the day before slaughtering hair samples were shaved from the left lumbar zones of six animals per treatment, chosen at random, in order to evaluate the concentration of cortisol as a stress marker. Dressing percentage and carcass classification were obtained from the slaughterhouse.

### 2.3. Chemical Analysis

Samples of the different feedstuffs, orts and faeces from the digestibility balances were ground in a hammer mill fitted with a 1 mm sieve and analysed following the procedures of AOAC [29] for DM (ref. 934.01), OM (ref. 942.05), crude protein (CP; ref. 976.05) and ether extract (EE; ref. 2003.05; only in feedstuffs). The concentration of NDF was analysed using an Ankom 200 Fiber Analyzer (Ankom Technology), as described by Mertens [30], using α-amilase and sodium sulphite, and results were expressed exclusive of residual ashes. Acid detergent fibre (ADF) and acid detergent lignin were analysed as described by AOAC ([29]; ref. 973.18) and Robertson and Van Soest [31], respectively (only in feedstuffs). The total starch content of the concentrates was determined enzymatically from samples ground to 0.5 mm using a commercial kit (Total Starch Assay Kit K-TSTA 07/11, Megazyme, Bray, Ireland). Alkanes in feedstuffs, refusals and faeces were determined following the procedures described by Keli et al. [32] for the analysis of faecal samples.

Gas samples from in vivo and in vitro trials were analysed for methane concentrations via a manual injection of 100 μL onto a 30 m × 0.530 mm HP-1 capillary column (methyl siloxane; 1.5 μm thickness) in an Agilent 6890 apparatus (Agilent Technologies España S.L., Madrid, Spain) fitted with an automatic injector and flame ionization detector. The split ratio was 20:1. The carrier gas was helium (12 mL/min), as was the make-up gas to the detector (25 mL/min). The injector was set to 200 °C, the oven to 120 °C, and the detector to 350 °C for the whole process. Peak area data were processed using the HP ChemStation software (version A.08.03).

Refusals in polyester bags were analysed for DM and starch content, whereas the concentration of Yb in samples of faeces was determined as described by Keli et al. [33]. Cortisol in hair was determined as in Moya et al. [34] using an ArborAssays kit (ArborAssays, Ann Arbor, MI, USA).

### 2.4. Mathematical and Statistical Methods

The estimation of intake in group- and loose-housed animals using *n*-alkanes as markers was performed as in Keli et al. [33], whereas the disappearance of DM and starch from polyester bags on time was fitted to the first-order kinetic equation described by Ørskov and McDonald [35]. The concentration of Yb in faeces was adjusted to a decreasing exponential function in which the slope represents the fractional rate of passage through the rumen [36]. The effective degradability of both DM and starch were calculated as in Ørskov and McDonald [35].

Data on final LW, ADG, potential and effective degradability of DM and starch and dressing percentage were analysed using the PROC MIXED of SAS (SAS Inst. Inc., Cary, NC, USA, v 9.4) with treatment as the fixed effect and animal as random. Values were corrected via covariance using the initial body weight and age as covariates, and three animals from treatment B had to be removed from the statistical analysis due to infectious processes after cannulation with the small fistulae. The intake of straw and concentrate during digestibility balances, and digestibility values, were also analysed using PROC MIXED, including treatment and balance as fixed effects and animal as random. Regression between straw and concentrate intake was performed using the PROC REG protocol of SAS. Digestibility values were covariated with OM intake, and one animal from the third balance had to be removed from the data set for the same reason as above. Rumen fermentation variables (including methane production) were analysed as repeated measures with the MIXED procedure with treatment, sampling time nested within day, sampling day and all possible interactions as fixed effects, and animal as random. Sampling time nested within day was used as a repeated measure. The in vitro production of gas and methane was assessed following a similar model with treatment, balance, sampling time and their interactions as fixed effects, and animal as random. Frequencies of the different carcass classifications were analysed using the PROC FREQ procedure of SAS. The variance–covariance structure was selected based on the lowest Akaike information criterion on all occasions. Cortisol in hair was analysed using PROC MIXED, with treatment as fixed effect. For all data, differences were considered significant if *p* < 0.05.

## 3. Results

### 3.1. Animal Performance

The treatment did not affect (*p* > 0.1) the final LW, ADG, dressing percentage (Table 2) or carcass classification. The most frequent categories, according to the SEUROP classification system (OJEC, [37], were AO + 2 (16.5%), AR-2 (13.2%) and AO2 (11.0%). In this classification, A stands for males between 12 and 24 months of age, O stands for not very good carcass conformation (medium muscular development), R stands for good carcass conformation (good muscular development) and + and − stand for higher or lower values within a category. Number 2 indicates minor subcutaneous fat content, with muscles always apparent.

### 3.2. Intake and Digestibility

Unfortunately, the intake results of the group- and loose-housed animals were abnormal (diet composition estimated with *n*-alkanes including 100% straw and no concentrate), and hence are not shown.

Concentrate and straw intake during the digestibility balances, expressed either as kg/d or g/kg LW^0.75^, did not vary between treatments or balances (*p* > 0.1), but the proportion of straw, expressed as a percentage of the total DM intake, was affected by the interaction between treatment and balance (*p* = 0.02) (Table 3). As such, animals fed diet M consumed more straw during the second and third balances, whereas in animals fed diet B the straw intake was higher during the first balance, intermediate during the second balance and higher during the third balance. Differences between treatments were only significant during the second balance when animals fed the M diet consumed a higher proportion of straw than animals with diet B.

The regression between straw and concentrate intake was not significant (kg straw/day = 0.0314 × kg concentrate/day + 0.5039; R^2^ = 0.0315; *p* > 0.1).

The digestibility of DM or its fractions was not influenced by OM intake, and only showed differences between treatments (*p* < 0.05) in the case of DM and OM (higher for diet M), and between balances (*p* < 0.05) for OM (lower values for the third balance, without differences between the first two). The interaction between treatments and balances was not significant (*p* > 0.05) in any case. The intake of digestible OM was not affected by either treatment or balance (*p* > 0.1).

### 3.3. Rumen Fermentation

The effect of the majority cereal was only significant for the concentration of lactic acid (*p* = 0.005), which was higher for the M treatment (Table 4). The sampling day also affected lactic acid concentrations (*p* < 0.05), and methane concentration in vivo and the molar proportion of acetate in the rumen (*p* < 0.05). The interaction between treatment and the day of sampling only affected (*p* = 0.003) VFA concentration, although when comparisons between treatments were made within each sampling day there was no significance. The sampling time within a day did not influence any of the variables of rumen fermentation (*p* > 0.1). Methane concentrations in samples obtained in vivo were very low with both M and B treatments, with no differences between them (*p* = 0.21).

The interaction between treatment and balance significantly affected (*p* = 0.03) the cumulative gas production in vitro. In this respect, differences between treatments were significant only when samples from the third balance were incubated, with diet B producing more gas than diet M (average daily values of 64.2 vs. 53.6 mL/g incubated OM). As expected, the time of incubation also had a significant effect (*p* = 0.02) on cumulative gas production. Methane concentration was only affected by the time of sampling (*p* = 0.003), being higher at 12 than at 6 h (1.40 vs. 0.83 mM/g incubated OM). Average values for the two treatments were 1.13 and 1.09 mM/g incubated OM for M and B, respectively (*p* = 0.29).

The average potential degradability of the concentrates DM in sacco was 83.3% for diet M and 91.4% for diet B, the being difference attributable to a higher ‘b’ fraction (non-soluble but potentially degradable) in the latter (47.4% vs. 54.8% for M and B, respectively; *p* < 0.05). As the rates of passage through the rumen did not differ between treatments (0.060 h^−1^ vs. 0.054 h^−1^ for M and B, respectively; *p* > 0.05), effective degradability was also not affected by treatment (58.3% vs. 64.1% for M and B, respectively; *p* > 0.05). The effective degradability of starch was not different between treatments (64.4% vs. 64,1% for M and B, respectively; *p* > 0.05).

### 3.4. Cortisol in Hair

The concentration of cortisol in hair, as an indicator of animal stress, did not differ among treatments (*p* = 0.31; 2.16 vs. 2.44 pg/mg hair for animals fed M and B, respectively).

## 4. Discussion

### 4.1. Intake, Digestibility and Animal Performance

Previous studies have found no differences in intake among groups in steers [21,22,23,38] or lambs [39] fed maize- or barley-based diets. It must be pointed out that the intake results of the present work were obtained in metabolism crates, and the question here is whether group- and loose-housed animals would have had similar intakes. As stated above, the *n*-alkanes technique was tried to estimate the intake of the animals used in the digestibility balances once back to their pens. However, the results were rather abnormal, as diet composition estimated with *n*-alkanes included 100% straw and no concentrate. This outcome may have arisen due to the very low faecal recovery of the different faecal alkanes. In fact, the highest recoveries were obtained for hentriacontane (C_31_; 43% on average for the three digestibility balances), tritriacontane (C_33_; 38% on average) and nonacosane (C_29_; 37% on average), but these values were well below those obtained in other papers dealing with beef cattle (93%, 75% and 78% for C_29_, C_31_ and C_33_ in Oliván et al. ([40], for example). An error due to the incomplete collection of faeces in the metabolic cages was discarded as digestibility values were well in accordance with other published results from experiments using beef cattle fed similar diets to those used in the present work [41]. The concentration of alkanes in barley straw (62, 100 and 25 mg/kg DM, on average, for C_29_, C_31_ and C_33_, respectively) was 4.5–13.2 ten-fold the concentrations in compound feeds (5.9, 7.6 and 5.6 mg/kg DM). As concentrates were constituted between 86.5% and 93.8% of the total ration (Table 3), a small likely analytical bias due to their low concentrations of *n*-alkanes would surely have had a dramatic effect on diet composition and intake estimations. Valiente et al. [42] validated the use of *n*-alkanes for the estimation of intake, digestibility and diet composition in sheep fed a mixed grain–roughage diet, but in their case the maximum proportion of concentrates was 60%. Even though no direct estimations of intake in group- and loose-housed animals were possible, the structure of the company allowed a more-or-less accurate knowledge of the amounts of concentrates put in the silos, as well as of the leftovers. From these data, an average intake of 8.3 and 11.4 kg DM/d was estimated for M and B calves, estimations 21% and 36% higher, respectively, than the average concentrate intake recorded during the balance trials. A similar drop in intake during the confinement in metabolism crates has already been published elsewhere [23], and has been attributed to the stress associated with the change in environment, as well as to the decreased energy expenditure due to the decreased activity of the cattle while restrained. No estimates of straw intake were available for group- and loose-housed animals.

Digestibility values obtained in the present experiment were similar to those previously published using similar animals and diets [23,41], and were not affected by the majority cereal in the concentrate, except for DM and OM (higher for diet M). However, differences were numerically low (75% vs. 71%, and 77% vs. 73% for DM and OM, respectively), and differences in digestible OM intake were not significant. As a result, ADG values, dressing percentage and carcass classification were not affected by treatment. Daily gain values were in accordance with previously published results obtained with similar animals and diets [41], as were dressing percentage figures for the Montbéliarde breed [43]. The carcass quality of our animals was worse than that of the bulls used in the experiment by Chládek et al. [43]. However, these latter authors worked with older animals (683 kg at slaughter) which had grown at lower ADG (1.14 kg/d).

### 4.2. Rumen Fermentation

None of the variables of rumen fermentation were affected by the time of sampling. This finding was rather surprising, as the samples of rumen liquor were obtained from the animals during the balance trials, where the animals were fed once daily in the morning. The only explanation we can speculate on is that the animals were used to continuous feeding in the barns, and that they maintained a continuous feeding behaviour while in the metabolism crates. Unfortunately, feeding behaviour during the digestibility balances was not studied in the present experiment; however, the fact that the pH values recorded in the different sampling days and hours was not below 6.0, except for the first day of sampling of the first digestibility balance (5.99, 5.86 and 5.68 at 0, 3 and 9 h after feeding), supports the idea of a steady intake to avoid digestive discomfort (a pH of 5.6 or below being considered the benchmark for ruminal acidosis; [44,45]. The lack of differences in rumen pH when comparing diets based on barley or maize has also been pointed out in dairy cows [46,47] or beef cattle [21,22,48]. As rumen pH may be a source of digestive discomfort, the lack of differences between treatments M and B is in accordance with the lack of differences in cortisol concentrations in hair as an indicator of stress.

Methane production depends on gas production and on its concentration. In the present work, gas production was not measured in vivo, but it was in vitro, and this latter method was designed for the evaluation of ruminant feeds [25,49]. The objective of the present paper was not to obtain actual values of methane production from feedlot cattle, but to check the effect of partially substituting maize with barley on different traits, including methane production. From this point of view, and due to the absence of differences in gas production in vitro, it can be concluded, in general, that the majority cereal in the concentrate (maize or barley) does not affect gas production in the rumen. The higher cumulative gas production in vitro with diet B during the third balance was difficult to explain, as this diet had more NDF than diet M, and the DM and OM digestibility in vivo were lower. The effective degradability of DM did not differ between M and B treatments, also, even though it was 10% numerically higher for the latter. However, the effective degradability of starch was very similar between treatments (64.4% vs. 64,1% for M and B, respectively).

The concentration of methane in samples obtained from the rumen of the animals was very low (0.38 and 0.85 mM for treatments M and B, respectively, equivalent to 0.93% and 2.07% of the total gas collected; Table 4), and was affected by the day of sampling (*p* < 0.05). On their part, the production of methane from in vitro incubations was much higher (0.83 vs. 1.40 mM/g incubated OM at 6 and 12 h of incubation). As the method of analysis was the same (gas chromatography) in the in vivo and in vitro trials, the only outcome was that the way gas samples were obtained from the animals in vivo was inadequate. In any case, the production of methane at 6 h of incubation in vitro (13.32 g/kg OM intake, on average) was close to the daily average obtained by Beauchemin and McGinn [23] in feedlot cattle using environmental chambers. In their case, there were differences (*p* < 0.05) between animals fed a maize diet (9.7 g/kg OM intake) or a barley diet (13.8 g/kg OM intake), but cereals represented 81.4% of the ration (on a DM basis; 66.5% in the present experiment). On the other hand, Moate et al. [15] carried out an experiment with dairy cows fed rations where cereal grains constituted approximately 47% of the diet, and the source of forage was alfalfa hay. These authors expected that any inherent difference in fermentative properties between maize and barley should have been evident in methane metrics. In this respect, Fellner et al. [19] conducted an in vitro study and reported that when substrates containing 48% maize or barley were fermented in continuous fermenters, the barley substrates yielded greater methane. However, Moate et al. [15] found no differences in vivo between the maize and the barley diets with respect to methane emissions or methane yields. This lack of differences was also evidenced in a recent experiment [50] which used the samples obtained by Moate et al. [50]. In our case, the proportion of cereals in the diet was ca. 60.5%, and hence we can speculate that in our conditions methane metrics were not affected by the source of grain, even in in vitro conditions.

The lactic acid concentration was also affected by the day of sampling (*p* < 0.05), this being higher in samples from the first digestibility balance, lower in samples from the third digestibility balance and intermediate in samples from the second digestibility balance (95.1, 71.0 and 47.8, mg/L for balance 1, 2 and 3, respectively). The only reason we can identify for this is the increasing proportion of straw in the diet (Table 3), although this was not significant as digestibility balances progressed (6.3%, 10.7% and 10.9% for balances 1, 2 and 3, respectively). The concentration of lactic acid was also affected by the majority cereal (*p* = 0.005), but in the opposite way to that expected. In fact, it was higher in M than in B animals (Table 4), which could be related to the higher NDF content in diet B (Table 1). Other papers have not found differences in lactic acid concentration when the majority cereal in the concentrate was maize or barley [21,22].

The lack of differences between maize- or barley-based concentrates on VFA concentration has already been reported [21,22,23,48]. On the other hand, Surber and Bowman [38] found a higher VFA production with barley than with maize in beef steers with the same DM intake between diets. For ammonia concentration, the lack of differences between maize- or barley-based concentrates supports the findings of Gimeno et al. [21]. The molar proportion of acetate was affected (*p* < 0.05) by the day of sampling (46.9%, 49.9% and 51.8% for balance 1, 2 and 3, respectively), which may have been for the same reason as the concentration of lactic acid: a non-significant increase in the proportion of straw in the diet.

## 5. Conclusions

A partial substitution of maize with barley in the concentrate given to intensively reared beef cattle does not affect methane yield in vitro in the incubated rumen liquid from animals consuming the same diets. In commercial conditions, the risk of acidosis is not increased when partially substituting maize with barley, and the animal performance is not affected. Hence, the above-mentioned strategy does not seem to reduce methane emissions from intensively reared beef cattle, and the inclusion of either cereal in greater proportions in the diet should be based only on their market availability and price.

## Figures and Tables

**Table 1 animals-13-03016-t001:** Ingredients and nutrient composition of the experimental diets.

	M	B	Barley Straw
Ingredients (as fed basis). g/kg			
Maize	455	155	
Barley	150	450	
Soy meal (440 g CP/ kg fresh matter)	88	74.8	
Sunflower meal	15	-	
Wheat middlings	50	-	
Maize gluten feed	95	150	
Soybean hulls	46	99.4	
Rapeseed meal	40	-	
Palm oil	33	40	
Urea	5.0	7.8	
Calcium phosphate	17.5	17.5	
NaCl	1.5	1.5	
Sodium bicarbonate	2.0	2.0	
Vitamin–mineral premix ^1^	2.0	2.0	
Nutrient composition (g/kg DM ± SEM; n = 3)			
OM	939 ± 2.7	937 ± 2.3	929 ± 10.9
CP	159 ± 1.8	167 ± 1.5	48.2 ± 7.62
EE	69.3 ± 1.55	61.5 ± 2.08	9.70 ± 3.899
Starch	396 ± 7.3	350 ± 12.2	-
NDF	198 ± 2.8	243 ± 7.2	764 ± 32.2
ADF	69.9 ± 2.09	93.6 ± 2.50	468 ± 23.0
Lignin	5.00 ± 0.346	5.23 ± 0.584	41.3 ± 2.57

M: control concentrate, consisting of maize as the majority cereal; B: treatment concentrate, consisting of barley as the majority cereal; DM: dry matter; OM: organic matter; CP: crude protein; EE: ether extract; NDF: neutral detergent fibre, ADF: acid detergent fibre; SEM: standard error of the mean; ^1^ vitamin–mineral premix declared composition (per kg): 3.5 × 10^6^ IU vitamin A, 0.75 × 10^6^ IU vitamin D_3_, 5 g vitamin E (alpha-tocopherol acetate), 2.5 g Fe sulphate monohydrate, 250 mg K-iodate, 250 mg Co acetate tetrahydrate, 1 g Cu sulphate pentahydrate, 15 g Mn oxide, 20 g Zn oxide, 100 mg Na selenite, 150 mg butylated hydroxytoluene and 260 g sepiolite.

**Table 2 animals-13-03016-t002:** Initial (ILW) and final live weights (FLW), average daily gain (ADG) and dressing percentage (DP) of feedlot beef cattle consuming different concentrates and barley straw.

Treatment (T)	M	B	SEM	*p*-Value
ILW (kg)	313	313	6.4	0.99
FLW (kg)	498	499	8.0	0.87
ADG (kg/d)	1.62	1.60	0.052	0.67
DP (%)	53.6	54.0	0.40	0.34

M: control concentrate, consisting of maize as the majority cereal; B: treatment concentrate, consisting of barley as the majority cereal; SEM: standard error of the mean; and *p*: probability of the differences.

**Table 3 animals-13-03016-t003:** Concentrate and straw intake (dry matter—DM) in metabolism crates, digestibility coefficients (g/100 g DM) of DM (DMD), organic matter (OMD), crude protein (CPD) and neutral detergent fibre (NDFD), and digestible organic matter intake (DOMI) in feedlot beef cattle consuming different concentrates and barley straw during three different digestibility balances.

Treatment (T)	M		B				SEM		*p*-Value	
Balance (Bal)	1	2	3	1	2	3		T	Bal	TxBal
Concentrate intake										
(kg/d)	7.91	5.97	6.60	7.59	8.44	8.96	1.074	0.15	0.83	0.38
(g/kg LW^0.75^)	88.7	65.3	71.0	87.8	93.9	97.14	11.98	0.13	0.76	0.43
Straw intake										
(kg/d)	0.52	0.93	0.86	0.52	0.72	1.04	0.172	0.94	0.12	0.58
(g/kg LW^0.75^)	5.78	10.17	9.25	6.02	7.92	11.24	1.795	1.00	0.13	0.55
(% of total DM intake/d)	6.18 a	13.47 bB	11.19 b	6.44 a	7.89 abA	10.53 b	0.813	0.02	0.002	0.02
DMD	74.99	75.98	75.25	72.87	72.36	67.65	0.781	0.004	0.051	0.070
OMD	76.21	77.06	76.40	76.12	73.91	69.25	0.912	0.015	0.049	0.059
CPD	76.57	72.09	72.26	74.96	74.49	68.67	1.984	0.58	0.12	0.41
NDFD	53.06	60.47	60.62	57.69	60.47	61.75	1.965	0.28	0.06	0.52
DOMI										
(kg/d)	6.00	4.99	5.35	5.76	6.39	6.50	0.885	0.34	0.96	0.61
(g/kg LW^0.75^)	67.3	54.6	57.5	66.6	71.0	70.4	9.75	0.29	0.90	0.65

M: control concentrate, consisting of maize as the majority cereal; B: treatment concentrate, consisting of barley as the majority cereal; LW: live weight; SEM: standard error of the mean; a, b: different letters within a row indicate differences between balances within a treatment at *p* < 0.05; and A, B: different letters within a row indicate differences between treatments within a balance at *p* < 0.05.

**Table 4 animals-13-03016-t004:** Average daily rumen pH, daily average concentrations of volatile fatty acids (VFA; mg/L), ammonia (mg/L), lactic acid (mg/L) and methane (CH_4_; mM) and daily average molar percentage (mmol/100 mmol) of the main VFA in the rumen fluid of feedlot beef cattle consuming different concentrates and barley straw.

Treatment (T)	M	B	SEM	*p*-Value
pH	6.23	6.32	0.097	0.51
Total VFA	89.1	81.7	3.47	0.14
Ammonia	100	130	15.6	0.23
Lactic acid	83.1	59.4	5.76	0.005
CH_4_	0.38	0.85	0.243	0.21
Acetate	49.2	49.9	2.22	0.82
Propionate	32.5	29.5	2.47	0.41
Butyrate	11.5	13.6	0.81	0.10
Iso-butyrate	1.41	1.47	0.141	0.79
Valerate	2.13	2.07	0.254	0.87
Iso-valerate	3.31	3.48	0.616	0.85

M: control concentrate, consisting of maize as the majority cereal; B: treatment concentrate, consisting of barley as the majority cereal; and SEM: standard error of the mean.

## Data Availability

The data presented in this study are available on request from the corresponding author.

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
