# Peer review of "Does Replacing Maize with Barley Affect the Animal Performance and Rumen Fermentation, including Methane Production, of Beef Cattle Fed High-Concentrate Diets On-Farm?"

_animals, 2023, doi:10.3390/ani13193016_

Round 1
Reviewer 1 Report
Dear author, It was a pleasure revising your manuscript. It’s not easy to conduct this kind of study and I know you made a tremendous effort to get these results regarding enteric methane emissions. Not all grains are equally effective in reducing methane production and studies regarding the effects of partial substitution of maize with barley on animal performance and rumen fermentation of beef calves intensively reared are needed.
I think your manuscript is well written and can be published with minimal revisions. You can improve the discussion and conclusion.
Abstract: Your abstract provided to your audience a clear summary of your study, methods, results, and conclusions. “Partial substitution of maize with barley in the concentrate offered to beef calves does not seem a promising strategy to decrease the emissions of enteric methane on-farm”
Introduction: The introduction of this manuscript conveys the basic information to the readers without obligating us to investigate previous publications and provided clues as to the gaps addressed by the present study.
Material and Methods; This section allows us to reproduce the study and the way that the data was analyzed is clear.
Discussions: The authors have interpreted and described the significance of their findings in relation to what was already known about the research problem being investigated and explained any new understanding or insights that emerged as a result of their research.
Please make sure the conclusion underscores the scientific value of your study/ highlight the applicability/limitation/future study instead of summarizing the manuscript.
The literature used to justify and interpret the results is too old. Please, update. Usually, at least 70% of the references are from papers published in the last 3 years.
Once again, I congratulate the author for this study and I look forward to read the revised version.
Cordially,
Author Response
Dear reviewer,
Thank you very much for your encouraging and supporting comments.
My answer to your point ‘The literature used to justify and interpret the results is too old. Please, update. Usually, at least 70% of the references are from papers published in the last 3 years’ is as follows: I have made a comprehensive search (Web of Science) of the relevant papers published on the topic of the effect of grain source on methane production in ruminants, and unfortunately I have found only four from the last years: three of them used an in vitro approach (Animals 2020, 10, 1316; doi:10.3390/ani10081316; Animals 2021, 11, 450. https://doi.org/ani11020450; J. Dairy Sci. 100:8881-8894; https://doi.org/10.3168/jds.2017-12675) and the fourth one was on dairy cows (PLoS ONE 17(5):e0268157; https://doi.org/10.1371/journal.pone.0268157). Only this latter has been included in the text (Discussion section) and in the references list because of its relevance. With respect to the use of n-alkanes as intake and digestibility markers in ruminants, no relevant papers have been published recently.
With respect to your second point ‘Please make sure the conclusion underscores the scientific value of your study/highlight the applicability/limitation/future study instead of summarizing the manuscript’, please note that the Conclusion has been changed.
Kind regards,
Antonio de Vega
Reviewer 2 Report
It was not clear why to replace corn with barley in the introduction or summary. Price? Market? Starch quality. Need to improve the discussion about starch, for example. Why would barley reduce methane?
Author Response
Dear reviewer,
Thank you very much for your kind and useful comments; please find my answers below.
Reviewer: It was not clear why to replace corn with barley in the introduction or summary. Price? Market? Starch quality. Need to improve the discussion about starch, for example. Why would barley reduce methane?
Author’s answer: In in vitro and in sacco studies, maize grain has been shown to have a lower rate and extent of degradation in the rumen than barley grain. A faster rate of rumen degradation of barley would decrease pH in the compartment, damaging the ability of methanogens to produce methane. Moreover, barley is usually (even though not always) cheaper than maize hence it would be a good alternative in terms of methane production and price. Most of these aspects have been included in the following paragraph of the introduction: ‘Not all grains are equally effective in reducing methane production in the rumen, and there are indications that the response of methane production to different proportions of grains may not be linear [14]. Moate et al. [15] found that dairy cows produced less methane when fed wheat than when fed maize or barley, and Herrera-Saldana et al. [16] observed in vitro that maize grain has a lower rate and extent of degradation than barley grain. This has also been confirmed in sacco [17,18] where dry matter (DM) and starch from barley had a faster degradation rate than DM and starch from maize. A faster rate of rumen degradation of barley would decrease pH in the compartment, damaging the ability of methanogens to produce methane’.
The discussion about starch has not been changed as it is well known that maize has a greater amount of starch than barley but with a lower degradation rate in the rumen. Total tract apparent digestibility is close to 100% in both cases.
Kind regards,
Antonio de Vega
Reviewer 3 Report
Comments to authors
Reducing methane emissions from ruminant production through the regulation of diet composition is a topic of importance and fall into the scope of the present journal. The authors aimed to examine the effects of partial substitution of maize with barley on animal performance and rumen fermentation of beef calves in farm conditions, and obtained some valuable research results.The experiment was well conducted, and the manuscript attracted a lot of attention, however, there was a small problem in the manuscript, the statistical analysis was inconsistent with the subsequent description of the results, and there was still ambiguity in the description. In addition, the following information provided in the manuscript could be improved.
Below are specific revisions :
L22 Change the semicolon behind the conditions and the examine to comma.
L98-100 It was noted that the beef cattles in each treatment were divided into two groups according to body weight, whether the eight fistula cows in each treatment group were evenly divided between the two weight groups.
L163 According to the experimental procedures, there are 12 cannula cattles and nine replicas were incubated per animal. So it will make a total of 108 bottles rather than 106 bottles.
L231-235 The description of this sentence is uncleare, were the infected animals cannulated with big or small cannula? If the animals cannulated with big fistulat were removed from the data set, the effective degradability of DM and starch couldn’t be statistic, as there is only two animals (cannulated with a semi-rigid cannula 95 mm i.d. and 200 mm long) for each treatment. Also delete the C behind treatment.
L236-237 Was only balance effect include in the statistical mode? If so the statistical model is wrong.
L240-241 Why the balance effect was considered in the statistic model of digestibility coefficient, while it was not considered in rumen fermentation?
L307-308 The description of this sentence is verbose in expression, rewrite. Also, please verify the unite of methane.
L311 Please list the b value of treatment M and B,respectively.
L313 Please verify the unite of the passage rate.
Table 4 Which statistical model was used for CH4 in in vivo experiment?
L385-386 The sentence is difficult to understant, rewrite.
L388 Delete in methane or change methane to gas.
L405 Change concentration to production.
L406 Please verify the unite of methane.
Reference
The abbreviation of the magazine name is irregular.
Author Response
Dear reviewer,
Thank you very much for your encouraging and useful comments; please find my answers below.
Reviewer: Reducing methane emissions from ruminant production through the regulation of diet composition is a topic of importance and fall into the scope of the present journal. The authors aimed to examine the effects of partial substitution of maize with barley on animal performance and rumen fermentation of beef calves in farm conditions, and obtained some valuable research results. The experiment was well conducted, and the manuscript attracted a lot of attention, however, there was a small problem in the manuscript, the statistical analysis was inconsistent with the subsequent description of the results, and there was still ambiguity in the description. In addition, the following information provided in the manuscript could be improved.
Author’s answer: The Results section has been reorganised somewhat to try and avoid ambiguity. The effects of sampling day and sampling time on rumen fermentation variables have been described in the text rather than in a table to avoid this latter being extremely complex. Also the results from in vitro incubation of the different diets have only been described in the text as there were only two variables measured. Same applies to potential and effective in sacco degradability of dry matter and starch, and to cortisol concentration in hair.
Reviewer: L22 Change the semicolon behind the conditions and the examine to comma.
Author’s answer: Changed.
Reviewer: L98-100 It was noted that the beef cattle in each treatment were divided into two groups according to body weight, whether the eight fistula cows in each treatment group were evenly divided between the two weight groups.
Author’s answer: Within each treatment, the animals whose weight were closer to the average were chosen for cannulation. This has been sated in the new version of the paper.
Reviewer: L163 According to the experimental procedures, there are 12 cannula cattle and nine replicas were incubated per animal. So it will make a total of 108 bottles rather than 106 bottles.
Author’s answer: Corrected.
Reviewer: L231-235 The description of this sentence is unclear, were the infected animals cannulated with big or small cannula? If the animals cannulated with big fistula were removed from the data set, the effective degradability of DM and starch couldn’t be statistic, as there is only two animals (cannulated with a semi-rigid cannula 95 mm i.d. and 200 mm long) for each treatment. Also delete the C behind treatment.
Author’s answer: The animals removed had been cannulated with the small fistula, and this has been stated in the new version of the paper. Treatment C has been substituted with treatment C.
Reviewer: L236-237 Was only balance effect include in the statistical mode? If so the statistical model is wrong.
Author’s answer: I meant that, apart from treatment and animal effects, intake and digestibility also included the balance effect. This has been clarified in the new version of the paper.
Reviewer: L240-241 Why the balance effect was considered in the statistic model of digestibility coefficient, while it was not considered in rumen fermentation?
Author’s answer: The model included day of sampling. Day 1 and 2 corresponded to the first balance, day 3 and 4 to the second balance, and day 5 and 6 to the third balance. So including day of sampling implicitly means the inclusion of the balance effect.
Reviewer: L307-308 The description of this sentence is verbose in expression, rewrite. Also, please verify the unite of methane.
Author’s answer: The sentence has been rewritten and the methane unit verified.
Reviewer: L311 Please list the b value of treatment M and B, respectively.
Author’s answer: Listed.
Reviewer: L313 Please verify the unite of the passage rate.
Author’s answer: Verified (proportion per time unit, hour in this case).
Reviewer: Table 4 Which statistical model was used for CH4 in in vivo experiment?
Author’s answer: The same as for the rest of rumen fermentation variables. This has been clarified in L 243 of the new version of the paper.
Reviewer: L385-386 The sentence is difficult to understand, rewrite.
Author’s answer: The sentence has been rewritten (L378 of the new version of the paper).
Reviewer: L388 Delete in methane or change methane to gas.
Author’s answer: ‘in methane’ has been deleted (L393 of the new version of the paper).
Reviewer: L405 Change concentration to production.
Author’s answer: Changed.
Reviewer: L406 Please verify the unite of methane.
Author’s answer: Verified. In the in vitro experiment methane production was expressed as mM/g incubated OM.
Reviewer: The abbreviation of the magazine name is irregular.
Author’s answer: All journal abbreviations have been checked according to Web of Science (https://images.webofknowledge.com/images/help/WOS/A_abrvjt.html) and corrected where appropriate.
Kind regards,
Antonio de Vega
Reviewer 4 Report
You do not mention the results of methane production in the Abstract, it is indispensable here.
I am puzzled about methane concentration in this study. Firstly, as mentioned in line 295-297, sampling day affected methane concentration in the rumen (P<0.05), thus the conclusions of partial substitution of maize with barley in the concentrate given to intensively reared beef cattle do not affect methane yield in vitro might not accurately. Secondly, it is necessary to indicate the data of sampling day affected methane concentration in the rumen (P<0.05), which one is higher and analyze the possible reasons? Thirdly, it is better to add a table to descript the gas production and methane concentration in vitro detailedly. Fourthly, it is very interesting if you can comparative analyze and discuss the data of methane concentration in vivo and in vitro.
Author Response
Dear reviewer,
Thank you very much for your valuable comments; please find my answers below.
Reviewer: You do not mention the results of methane production in the Abstract, it is indispensable here.
Author’s answer: Yes, of course. A mention to the results of methane production has been included in the Abstract.
Reviewer: I am puzzled about methane concentration in this study. Firstly, as mentioned in line 295-297, sampling day affected methane concentration in the rumen (P<0.05), thus the conclusions of partial substitution of maize with barley in the concentrate given to intensively reared beef cattle do not affect methane yield in vitro might not accurately.
Author’s answer: Sampling day affected methane concentration in samples obtained in vivo, whereas sampling time affected the concentration of methane in samples from the in vitro trial. We have tried to clarify this aspect in the revised version of the paper.
Reviewer: Secondly, it is necessary to indicate the data of sampling day affected methane concentration in the rumen (P<0.05), which one is higher and analyze the possible reasons?
Author’s answer: The ranking was as follows (in mM, as shown in Table 4): Day 1-4.69, Day 2-2.17; Day 3-0.26; Day 4-0.25; Day 5-0.96; and Day 6-0.67. Day 1 and 2 corresponded to the first digestibility balance; Day 3 and 4 to the second balance; and Day 5 and 6 to the third balance. Looking at Table 3, there are no reasons to blame differences in methane concentration between days on differences in straw and concentrate intake. Anyway, methane concentrations in samples from the in vivo trial were unrealistically low, and hence are not discussed.
Reviewer: Thirdly, it is better to add a table to descript the gas production and methane concentration in vitro detailedly.
Author’s answer: This would result in a 2 (treatments) x 2 (sampling times) table which would be rather short. Average values for the two treatments were 1.13 and 1.09 mM/g incubated OM for M and B, respectively (P=0.29). This information has been included in L315-316 of the revised paper.
Reviewer: Fourthly, it is very interesting if you can comparative analyze and discuss the data of methane concentration in vivo and in vitro.
Author’s answer: As stated above, the results from the in vivo trial were unrealistically low, hence we decided not to discuss them (Please see L 408-414 of the new version of the paper).
Kind regards,
Antonio de Vega
Round 2
Reviewer 1 Report
Dear authors,
I am fine with the revised version of your manuscript.
You have incorporated my suggestions.
Cordially,
Abmael
Reviewer 4 Report
The authors have improved the manuscript, it can be accepted in present form.